# Online and Face-to-Face Mat Pilates Training for Long COVID-19 Patients: A Randomized Controlled Trial on Health Outcomes

**DOI:** 10.3390/ijerph21101385

**Published:** 2024-10-19

**Authors:** Ana Clara Ribeiro Cunha, Juliana Cristina Silva, Caroline Pereira Garcês, Tássia Magnabosco Sisconeto, João Luiz Rezende Nascimento, Ana Luiza Amaral, Thulio Marquez Cunha, Igor Moraes Mariano, Guilherme Morais Puga

**Affiliations:** 1Exercise, Women and Cardiometabolic Health Research Group, Faculty of Physical Education and Physical Therapy, Federal University of Uberlândia, Uberlândia 38400-678, MG, Brazil; anaclararibeirocunha1@gmail.com (A.C.R.C.); julianasilvacristina@yahoo.com.br (J.C.S.); carolgarces@outlook.com (C.P.G.); tassiamagnabosco@hotmail.com (T.M.S.); jolurenas@ufu.br (J.L.R.N.); anaribeiro.am@gmail.com (A.L.A.); igormmariano@gmail.com (I.M.M.); 2School of Medicine, Federal University of Uberlândia, Uberlândia 38400-902, MG, Brazil; thcunha@yahoo.com.br

**Keywords:** online exercise, fatigue, Pilates, functional capacity, body composition, SARS-CoV-2

## Abstract

This study investigated the impacts of online and face-to-face Mat Pilates training in adults with persistent symptoms of long COVID on health outcomes. Forty-nine patients (52 ± 5.85 yr.) diagnosed with long COVID related to fatigue symptoms were randomly included in three groups: online Mat Pilates training (n = 16), face-to-face Mat Pilates training (n = 15), and a control group (n = 18) without training. Mat Pilates training was conducted three times a week for 12 weeks. Fatigue, functional capacity, anthropometrics, body composition, and cardiometabolic markers were assessed before and after the interventions. Two-factor Generalized Estimating Equation analyses identified significant differences with Bonferroni post hoc testing (*p* < 0.05). After the intervention, only the face-to-face Mat Pilates training group had an improved total, physical and mental fatigue, trunk isometric strength, upper limb muscle endurance strength, and aerobic capacity (*p* < 0.05). No changes were found in fat mass, muscle mass, free fat mass, % of fat, body mass, body mass index, or waist and hip circumferences. No significant changes were observed in blood glucose, glycated hemoglobin, triglycerides, total cholesterol, high-density lipoprotein, low-density lipoprotein, or blood pressure (*p* > 0.05). Our results highlight the potential of face-to-face Mat Pilates training as an effective intervention to mitigate persistent symptoms of long COVID related to fatigue and functional capacities.

## 1. Introduction

SARS-CoV-2 infection can lead to signs, symptoms, and conditions that may emerge or endure past the acute phase. Symptoms persisting beyond 12 weeks are commonly referred to as post-COVID-19 condition, post-acute COVID-19 syndrome, or recently as long COVID [1]. This condition may involve multiple organ systems, including the cardiovascular, nervous, hepatic, and renal systems, and can present with a relapsing–remitting pattern, potentially progressing or worsening over time [2,3,4].

The primary manifestations of long COVID include tiredness, fatigue, cognitive impairment, shortness of breath or dyspnea, headache, chest pain, coughing, musculoskeletal pain, depression or anxiety, and anosmia or ageusia [5,6]. Currently, approximately 40% of COVID-19 patients progress to a persistent phase of clinical manifestations, with fatigue being one of the most prevalent symptoms [5].

Regular exercise has been associated with a significant reduction in the chances of hospitalization due to COVID-19, demonstrating its essential role in promoting health and preventing severe complications related to the infection [7]. Low- to moderate-intensity exercises, in particular, can be important in contributing to the reduction in the risk of developing more severe forms of COVID-19 [8]. Such exercise may also be an important strategy for treating long COVID symptoms, such as muscle weakness, physical limitations, and fatigue. Additionally, it can protect against functional limitations that may be caused by the worsening of the syndrome [9,10]. It is described that structured exercise programs lead to improvements in cardiopulmonary fitness and functional status [11], and may be beneficial for patient post-hospitalization and with persistent symptoms, that is, long COVID [5]. Among the different types of exercises proven to be beneficial to health and used in the prevention and treatment of diseases, the Pilates method has been recommended [12,13].

The Pilates method is a dynamic technique that emphasizes strength, flexibility, and balance [14]. The literature has demonstrated that Pilates training promotes improvements in skeletal muscle properties, mental health, and physical fitness [15], and it is effective in enhancing functional capacity and overall health conditions. This modality boasts a high adherence among the population [16], so it may improve physical capacities limited by long COVID [17,18,19,20,21], and it can also enhance quality of life in these patients [22].

The recent experience of the COVID-19 pandemic gave healthcare professionals the opportunity to disseminate online training methods [23,24]. Telerehabilitation is an evolving approach that diminishes obstacles to services, such as time, cost, and distance. It eliminates financial burdens, promotes cost savings, and facilitates the interaction between individuals and healthcare professionals within a virtual environment [25]. There are indications that, in healthy individuals, online Mat Pilates training has positive effects on trunk proprioception and central muscle endurance [26], reducing weight, depression [27], and other health outcomes [25]. Thus, Mat Pilates proved to be a viable option for practice during the pandemic period [25]. As there are few studies investigating the effects of face-to-face and online exercises, more studies are needed to elucidate the effects of this exercise training in long COVID patients, especially on health outcomes such as functional capacity and cardiovascular and metabolic indexes.

Therefore, recognizing the need for well-designed and controlled studies, we investigated the impacts of online versus face-to-face Mat Pilates in adult patients with persistent symptoms of long COVID relating to fatigue, functional capacity, body composition, and cardiometabolic markers of health. We hypothesized that both face-to-face and online Mat Pilates exercise training would improve overall health outcomes and fatigue related to the long COVID condition.

## 2. Materials and Methods

### 2.1. Study Design

This was a controlled, randomized, parallel-group clinical trial carried out in accordance with the Declaration of Helsinki. The project was registered in the Brazilian Clinical Trials Registry (Rebec)—UNT: U1111-1292-5013. The manuscript was described following the guidelines of the Consolidated Standards of Reporting Trials (CONSORT) [28].

### 2.2. Randomization

Randomization was performed 1:1, with the sample divided into blocks of 9 individuals using the website http://www.jerrydallal.com/random/random_block_size_r.htm (accessed on 14 October 2022). The process was carried out by an independent researcher who had no involvement with the participants.

### 2.3. Participants

The assessments were conducted at the Cardiorespiratory and Metabolic Physiology Laboratory (LAFICAM) at the Federal University of Uberlândia (UFU). This study was advertised on social media, and participants who expressed interest contacted us from October 2022 to August 2023. Eligibility criteria included patients aged 18 to 60 years who had COVID-19 infection confirmed by viral tests including nucleic acid amplification tests, PCR tests, or antigen tests. They also had to receive a diagnosis of COVID-19 at least 3 months prior, present COVID-related fatigue, and meet the current diagnostic criteria for long COVID [29], with the presence of fatigue as determined using the Fatigue Assessment Scale (FAS) [30]. Participants with any functional limitation or without medical clearance for exercise were excluded. The entire study was explained to the volunteers, and all signed the free and informed consent form. Some participants were taking medication for cardiometabolic diseases such as hypertension, diabetes, dyslipidemia, and thyroid disorders. They also had some comorbidities that were evaluated based on the anamnesis and medicaments that the participants reported, and also based on the physical exams that they underwent at the beginning of this study. All these pieces of information are presented in Table 1.

### 2.4. Outcomes

Fatigue scales, functional capacity tests, physical activity level assessments, and functional scale evaluations were performed and fasting blood samples were collected at the baseline and after the 12-week intervention by the same researchers, at the same facilities. The test evaluators were unaware of which group each participant belonged to.

### 2.5. Fatigue Assessment

Fatigue was assessed using the Fatigue Assessment Scale (FAS) [30,31], which comprises ten questions to assess fatigue, with five questions focused on physical fatigue and five on mental fatigue. The classification is based on the sum of the responses to the questions, with scores of 10–21 indicating “no fatigue”, 22–34 indicating the “presence of fatigue”, and scores greater than 35 classified as “extreme fatigue”.

### 2.6. Physical Activity Level

The participants’ physical activity level was assessed for characterization purposes, using the Brazilian Portuguese version of the International Physical Activity Questionnaire (IPAQ) [32].

### 2.7. Functional Status/Capacity

The Post-COVID-19 Functional State Scale was administered [33], a questionnaire used to classify the patient’s functional state on a scale from 0 to 4, indicating the severity of their limitations (the higher the score, the greater the functional limitation).

### 2.8. Functional Capacity Assessments

Functional capacity was evaluated by the timed up-and-go test (TUG) [34], upper limb endurance [35], flexibility [35], handgrip isometric strength [36], trunk flexion and extension isometric strength [37,38,39], and a 6 min walk test [40].

### 2.9. Anthropometry and Body Composition

Body composition assessment was performed using Bioelectrical Impedance Analysis with the InBody^®^ 230 device (Seoul, South Korea). We measured body weight using a electronic scale (Filizola^®^, São Paulo, Brazil) and height using a fixed stadiometer (Sanny^®^, São Bernardo do Campo, Brazil). The circumferences of the hips (maximum circumference over the buttocks) [41] and waist (midway between the lower rib margin and the iliac crest) [42] were determined using a 0.5 cm wide inelastic measuring tape (Filizola^®^, São Paulo, Brazil).

### 2.10. Blood Pressure

For the measurement of resting blood pressure (BP), the participant remained seated and at complete rest for 10 min. BP was monitored before and after training using calibrated and validated automatic monitors [43] from the Omron^®^ brand (Kyoto, Japan), model HEM-7113. At each measurement point, three readings were taken, and the mean was used for analysis. Readings outside the 95% confidence interval were discarded, and the mean of the remaining two was considered.

### 2.11. Blood Analyses

We collected 15 mL blood samples after an overnight fast, 5 days before and 72 hours after the last exercise session, to eliminate the acute effects of the exercise. These samples were placed in EDTA or serum tubes with separator gel (depending on the exam) and then centrifuged at 3000 rpm for 15 min and stored in microtubes for future analysis. Concentrations of total cholesterol, triglycerides, high-density lipoprotein (HDL) and low-density lipoprotein (LDL) cholesterol, and glucose were determined by enzymatic colorimetric methods using commercial kits (Labtest, Lagoa Santa, MG, Brazil). The concentration of glycated hemoglobin was determined by the turbidimetry method. All analyses were performed using an automated system (Cobas Mira, Roche Instruments Inc., Bellport, NY, USA), except for adiponectin, which was determined by the human ELISA kit (ELABSCIENCE, Wuhan, China).

### 2.12. Intervention and Control

The intervention consisted of a Mat Pilates training program led by a certified instructor, conducted at the gymnasium of the Faculty of Physical Education at UFU over a period of 12 weeks. The program comprised 36 sessions, held three times per week. Each session lasted 50 min, divided into 5 min of warm-up, 40 min of core exercises, and 5 min of cool-down. A 45 s interval between exercises was stipulated to ensure adequate muscle recovery.

Prior to the commencement of the main program, a one-week familiarization phase was conducted to introduce participants to the fundamental principles of the Pilates method, such as breathing, concentration, centering, control, and fluidity. Additionally, during this week, exercises that would be performed throughout the training were demonstrated. This period was essential to ensure that participants were adequately prepared for the subsequent exercise practices.

For the training program, 26 classic exercises from the method developed by Joseph Pilates were selected, focusing on the activation of major muscle groups and the performance of multi-joint exercises (see Appendix A). The program was divided into two distinct workouts: Workout A and Workout B. Each workout consisted of 10 different exercises performed solely with body weight. Additionally, the program included 3 warm-up exercises and 3 cool-down exercises. Workouts A and B were alternated on training days.

The training program was divided into three difficulty levels, corresponding to 4 weeks of the program: Beginner, Intermediate, and Advanced. All levels started with 10 repetitions in the first two weeks and ended up with 12 repetitions. The beginner and intermediate-level exercises were developed by the researcher to prepare the volunteers for the correct execution of advanced-level exercises, which include Joseph Pilates’ basic exercises. These adaptations were simplified versions of the advanced exercises.

During the training, participants received detailed instructions on proper breathing techniques and body control. This guidance was crucial for ensuring the correct execution of the exercises. Additionally, the intensity of the training was monitored at the end of each session using the Borg Rating of Perceived Exertion Scale.

Adherence to the program was rigorously monitored, resulting in an 85% attendance rate, as verified through attendance records. In the online group, Pilates classes were recorded by instructors who provided explanations of Pilates principles and demonstrated correct exercise execution, adhering to the same training protocol as the face-to-face group. Each recorded session lasted 50 min and included a thorough demonstration of all exercises. These videos were shared three times a week via social media throughout the 12-week intervention. Additionally, the instructor documented attendance at the end of each day. The control group received a booklet of stretching instructions and was advised to perform them three times a week.

Figure 1 shows the flowchart. During the study, there were 21 losses, but these were not related to the training protocol.

### 2.13. Statistical Analyses

The necessary sample size was calculated a priori considering trunk flexion as the primary outcome [25], and this was analyzed on GLIMMPSE, adopting a significance level of 0.05 and a power of 80% and using a two-way analysis with two repeated measures for group-by-time interaction. Thus, a minimum sample size of 9 individuals was determined. However, given the principle of data conservatism, we considered the possibility of having an effect size smaller than that of the base study; therefore, we aimed to include at least 15 volunteers per group.

We employed Generalized Estimating Equations (GEEs) to analyze the longitudinal (pre to post) changes over time in three groups in all functional capacity, body composition, and blood analysis variables. The GEE model included intervention groups (face-to-face, online, and control), time (pre and post 12 weeks), and their interaction (group*time). A post hoc test with Bonferroni adjustment was performed when significant F values were found. The adequacy of the GEE models was assessed using diagnostic procedures such as the quasi-likelihood under the independence model criterion (QIC) and examination of residuals. Data are presented as the mean ± SD for continuous variables or as the frequency and percentage for categorical variables. The fatigue assessment data are presented in a boxplot, providing a clear visualization of the distribution, median, interquartile range, and possible outliers of the fatigue scores among the different groups. Furthermore, we calculated the effect size of each group after the interventions using Cohen’s *d* [44]. The significance level was set at *p* ≤ 0.05. The multiple imputation technique was used in the intention-to-treat analysis for missing data. All analyses were performed on SPSS software (v.22.0).

## 3. Results

Table 1 presents the clinical and general characteristics of the volunteers and the FAS classification. All participants exhibited fatigue (control n = 7 (39%); online Mat Pilates n = 12 (75%); and face-to-face Mat Pilates n = 8 (61%)) or extreme fatigue (control n = 11 (61%); online Mat Pilates n = 4 (25%); and face-to-face Mat Pilates n = 5 (38%)), meeting the criteria for being diagnosed with long COVID [29]. There was no difference (*p* > 0.05) in age, weight, height, or BMI among groups. The volunteers were infected at least 3 months before participating in this project. There were no differences between the groups at the baseline. None of the participants had an absence of limitations. The number of comorbidities was evaluated based on the anamnesis and medicaments that the participants reported, and also based on the physical exams that they underwent at the beginning of this study.

Table 2 shows the anthropometric and body composition of the participants. No significant interaction (group × time) was found among groups.

Table 3 shows the functional capacity test results for the participants. The GEE analysis showed that only the face-to-face Mat Pilates group improved (*p* < 0.05) in endurance strength (16 ± 4 to 23 ± 4 repetitions), trunk flexion isometric strength (06.6 ± 3.0 to 12.7 ± 6.6 kg), and the 6 min walk test (463 ± 73 to 514 ± 39 m), with no differences in the other groups.

Table 4 shows the fasting blood test analysis and resting blood pressure values. Both systolic (114 ± 13 to 122 ± 12 mmHg) and diastolic (75 ± 9 to 81 ± 8 mmHg) blood pressures increased in the control group over time, with no difference in the Pilates exercise training groups.

According to the evaluation of the FAS (Figure 2), only the face-to-face Mat Pilates group showed a significant reduction in the total (33.5 ± 5.5 to 26.7 ± 7.1, *p* < 0.01), physical (18.1 ± 3.0 to 14.8 ± 4.2, *p* < 0.01), and mental (15.4 ± 3.5 to 11.8 ± 3.8, *p* < 0.01) score aspects of the scale. No difference was found in other groups over time.

## 4. Discussion

Our study investigated the impacts of online and face-to-face Mat Pilates training in adult patients with persistent symptoms of long COVID on fatigue, functional capacity, anthropometrics, body composition, and cardiometabolic markers. Our main findings were that only face-to-face Mat Pilates training improved physical and mental fatigue, muscular strength, and aerobic capacity. These findings show the effectiveness of Mat Pilates as a therapeutic approach for persistent symptoms of COVID-19.

Regular physical activity is correlated with a notable decrease in the risk of developing long COVID and a lower propensity for specific symptoms, including fatigue, neurological complications, and headaches [45].

The evaluation of post-COVID-19 functional capacities should include tests of strength and cardiopulmonary endurance [46]. Fatigue and muscle pain are commonly reported symptoms by long COVID patients, often associated with reduced levels of physical activity [6,47]. Previous studies [15,48] have highlighted the benefits of exercise rehabilitation in improving fatigue, functional capacity, and muscular strength. Our results corroborate these findings, demonstrating significant improvements in these capacities after face-to-face Mat Pilates training. Mat Pilates is proposed as a viable intervention for managing persistent COVID-19 symptoms [25]. We found improvements in a variety of tests, including TUG, upper limb endurance, trunk flexion isometric endurance, and the 6 min walk test. Moreover, this exercise training was able to improve fatigue symptoms that persist after COVID-19 infection. These results indicate that face-to-face Mat Pilates emerges as an effective strategy for enhancing functional capacity in individuals with long COVID.

On the other hand, online Mat Pilates training did not show beneficial results, contradicting our initial hypothesis. Recent studies showed that online Pilates training was able to improve muscular endurance, depression, and quality of life in healthy adults [25], as well as improve trunk proprioception and core muscle endurance in healthy volunteers [26], and reduce weight and depression in pregnant women [27]. Although these studies were conducted during the COVID-19 pandemic, none of them focused on patients with a long COVID diagnosis and persistent symptoms related to fatigue. Our results suggest that online Mat Pilates training may not be a viable option to improve health outcomes in these patients, especially after the end of lockdown and social distancing.

In studies where online training yielded positive results [25,26,27], the sessions were conducted synchronously, allowing real-time interactions between participants and instructors. However, in the present study, we opted for an asynchronous format, which may have contributed to the lack of significant results. Another possible explanation could be the post-pandemic context of our study. During the peak of the pandemic, stricter social isolation measures may have motivated greater dedication from volunteers. As restrictions eased over time, participants’ commitment to online training may have decreased, potentially influencing the observed outcomes. These findings align with previous studies suggesting that face-to-face training may be more effective than online training in promoting improvements in strength, physical capacities, and reduction in fatigue [49]. This underscores the importance of the face-to-face environment and direct supervision in optimizing the outcomes of Pilates training.

The lack of positive results in body composition, anthropometric parameters, and cardiometabolic parameters, both for online and face-to-face Mat Pilates training, corroborates previous studies [49,50]. This exercise may involve an insufficient exercise intensity, volume, and overall physiological stress to produce the necessary energy expenditure and cardiometabolic adaptation. Mat Pilates involves local muscle contraction and depends on good movement skills and knowledge if it is to be performed at a higher intensity. In that sense, we propose that it is very difficult in related research studies to achieve a sufficiently high heart rate, oxygen consumption, and energy expenditure in order to be able to improve fat oxidation, muscle mass, or cardiometabolic risk factors of health, such as blood glucose and the blood lipid profile. We did not find positive results with Mat Pilates training in this cohort of patients with long COVID, but some studies showed a beneficial effect on cardiovascular parameters after the same Mat Pilates exercise training was implemented in post-menopausal women [51].

Regarding the potential clinical applications of this study, Mat Pilates is a type of exercise with a high adherence among the population [16]. Face-to-face Mat Pilates has proven to be useful for treating important clinical parameters in patients with long COVID symptoms, such as physical and mental fatigue, lack of muscle strength, and limited aerobic capacity. The training method used in this study allows for the treatment of many patients simultaneously, compared to individual sessions, thereby optimizing treatment at a low cost. Furthermore, our findings highlighted the importance of the face-to-face involvement of exercise professionals in the treatment of long COVID. Moreover, they showed that understanding the physiological and clinical responses to the Pilates method in long COVID patients is crucial when considering the best training approach for this population, given the scarcity and/or low methodological quality of previous studies.

The strengths of this study were the well-designed protocol, following the rules of a randomized clinical trial, and using online and face-to-face exercise interventions compared to a control group. Additionally, the Mat Pilates exercises and training intervals were tailored according to participants’ physical fitness and experience in exercise performance, with the workload stepping up in increments over the 12 weeks. Although all participants met the current strict criteria for a long COVID diagnosis, we do not yet know much about long COVID’s presentation and consequences in the long term. Long COVID is still a new condition, and we need to know more about it and the expected duration of the symptoms and consequences over a long period of time.

## 5. Conclusions

In conclusion, the results of this study highlight the potential of face-to-face but not online Mat Pilates exercises training for 12 weeks as an effective intervention to reduce persistent symptoms of long COVID related to fatigue and functional capacities. The improvements in physical and mental fatigue, along with an increased functional capacity in terms of muscular strength and aerobic capacity, suggest that Mat Pilates can play a crucial role in the rehabilitation and recovery process for individuals affected by this condition. Nonetheless, our study had some limitations, such as the heterogeneous presentation of non-communicable diseases in the participants, the participants’ effort not being monitored in the online group, and the lack of standardized vaccination timing. Further research is needed involving different types of exercise in patients with long COVID to assess improvements in the associated symptoms.

## Figures and Tables

**Figure 1 ijerph-21-01385-f001:**
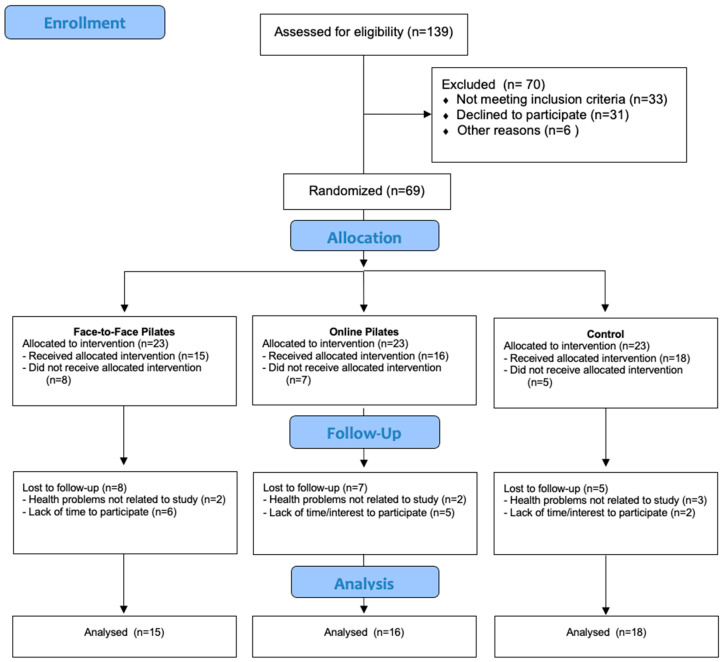
Flow chart of study process.

**Figure 2 ijerph-21-01385-f002:**
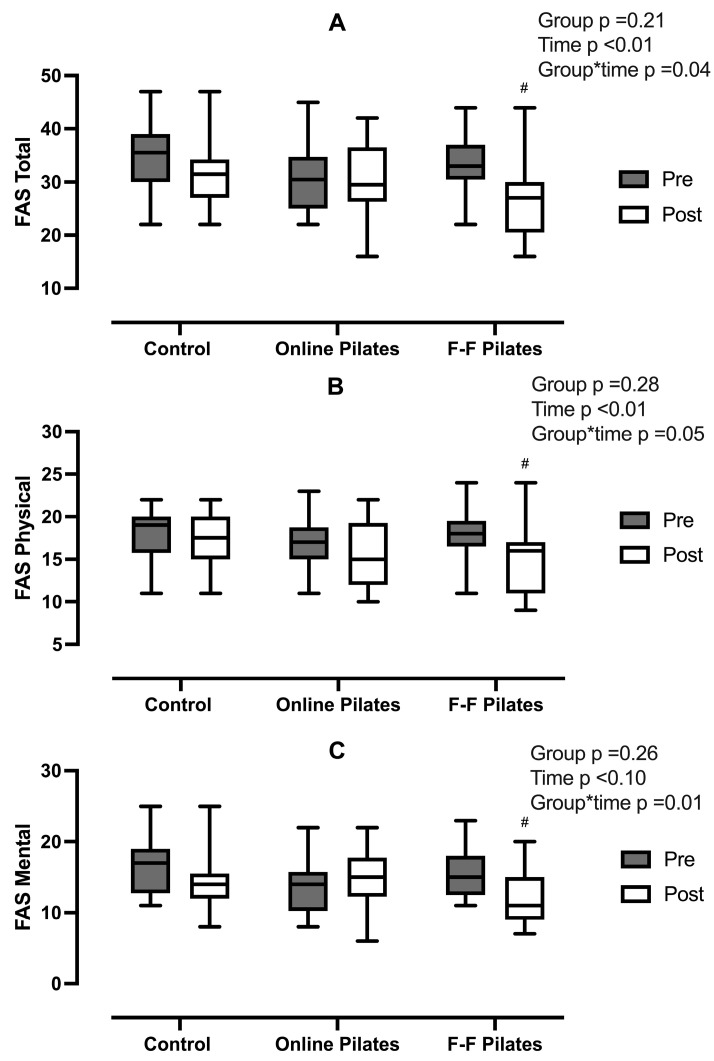
Fatigue Assessment Scale (FAS) results from total (Pannel (**A**)), physical (Pannel (**B**)), and mental scores (Pannel (**C**)) of volunteers allocated to control (n = 18), online pilates (n = 16), and face-to-face Pilates (n = 15) groups.

**Table 1 ijerph-21-01385-t001:** Baseline general characteristics of volunteers.

	Control Mean ± SD n = 18	Online Mean ± SD n = 16	Face to face Mean ± SD n = 15
Age, years	53 ± 6	53 ± 6	50 ± 6
Women, n (%)	17 (94)	15 (94)	12 (80)
Weight, kg	77.8 ± 26.6	82.9 ± 18.2	85.1 ± 27.1
Height, cm	160 ± 0.07	166 ± 0.15	163 ± 0.07
BMI, kg/m^2^	30.7 ± 11.2	32.2 ± 6.8	31.9 ± 8.3
Physical activity level—IPAQ			
Inactive, n (%)	3 (17)	0 (0)	1 (6)
Irregularly active, n (%)	7 (39)	11 (69)	8 (53)
Active, n (%)	7 (39)	4 (25)	6 (40)
Very active, n (%)	1 (6)	1 (6)	0 (0)
SpO_2_, %	96.9 ± 1.2	95.4 ± 2.8	96.2 ± 1.3
Systolic blood pressure, mmHg	114 ± 13	118 ± 13	112 ± 13
Diastolic blood pressure, mmHg	75 ± 9	76 ± 9	76 ± 9
Comorbidities			
Diabetic, n (%)	3 (16)	2 (12)	1 (6)
Hypertensive, n (%)	8 (44)	5 (31)	6 (40)
Dyslipidemia, n (%)	2 (11)	6 (37)	5 (33)
Others, n (%)	11 (61)	5 (31)	4 (26)
Medications			
Thiazide diuretic, n (%)	2 (11)	2 (12)	4 (26)
Beta blocker, n (%)	2 (11)	2 (12)	2 (13)
ACE I, n (%)	2 (11)	2 (12)	1 (6)
ARA II, n (%)	5 (28)	1 (6)	3 (20)
Antidiabetics, n (%)	4 (22)	4 (25)	1 (6)
Antidepressant, n (%)	3 (16)	5 (31)	2 (13)
Thyroid hormones, n (%)	5 (28)	1 (6)	1 (6)
Others, n (%)	11 (61)	9 (56)	6 (40)
Fatigue rating			
No fatigue, n (%)	0 (0)	0 (0)	0 (0)
Fatigue, n (%)	7 (39)	12 (75)	8 (61)
Extreme fatigue, n (%)	11 (61)	4 (25)	5 (38)
Functional scale classification			
1, n (%)	1 (6)	1 (6)	0 (0)
2, n (%)	3 (17)	3 (19)	4 (26)
3, n (%)	8 (44)	11 (69)	7 (46)
4, n (%)	6 (33)	1 (6)	4 (26)

Data expressed as mean ± SD or frequency and percentage (%). SD, standard deviation; BMI, body mass index; SpO_2,_ peripheral oxygen saturation; ACE I, angiotensin-converting enzyme inhibitors; ARA II, angiotensin receptor antagonist II.

**Table 2 ijerph-21-01385-t002:** Anthropometry and body composition of volunteers allocated to control (n = 18), online Pilates (n = 16), and face-to-face Pilates (n = 15) groups.

	Pre	Post	*p* Group	*p* Time	*p* Group × Time	Power	Cohen’s *d*
Body Mass (kg)							
Control	77.8 ± 26.6	82.8 ± 17.4	0.92	0.93	0.32	0.094	0.22
Online Pilates	82.9 ± 18.2	82.0 ± 17.2					0.05
F-F Pilates	85.1 ± 27.1	80.4 ± 21.5					0.18
Body Mass Index (kg/m^2^)							
Control	30.7 ± 11.3	33.2 ± 07.0	0.92	0.61	0.32	0.101	0.28
Online Pilates	32.2 ± 06.7	32.3 ± 06.5					0.03
F-F Pilates	31.9 ± 08.3	30.7 ± 06.7					0.17
Waist Circumference (cm)							
Control	97.9 ± 23.2	101.2 ± 09.5	0.31	0.61	0.31	0.080	0.20
Online Pilates	105.4 ± 16.3	108.3 ± 14.0					0.20
F-F Pilates	105.3 ± 14.7	102.3 ± 09.5					0.23
Hip Circumference (cm)							
Control	112.4 ± 18.7	112.9 ± 09.7	0.41	0.62	0.04	0.247	0.03
Online Pilates	114.9 ± 12.8	122.7 ± 20.7					0.45
F-F Pilates	116.7 ± 17.1	111.7 ± 15.1					0.30
Fat Mass (kg)							
Control	36.8 ± 13.9	37.6 ± 12.6	0.72	0.20	0.56	0.054	0.05
Online Pilates	34.4 ± 13.2	36.2 ± 12.7					0.13
F-F Pilates	33.1 ± 14.3	33.3 ± 14.9					0.00
Muscle Mass (kg)							
Control	43.9 ± 6.7	43.4 ± 6.0	0.80	0.84	0.30	0.058	0.08
Online Pilates	42.6 ± 6.0	43.2 ± 5.6					0.11
F-F Pilates	44.8 ± 8.7	44.4 ± 8.2					0.04
Free Fat Mass (kg)							
Control	46.6 ± 7.2	46.0 ± 6.2	0.78	0.82	0.32	0.058	0.08
Online Pilates	45.2 ± 6.3	45.8 ± 5.8					0.10
F-F Pilates	47.6 ± 9.1	47.2 ± 8.7					0.04
%Fat (kg)							
Control	43.0 ± 7.8	43.1 ± 6.3	0.38	0.49	0.80	0.053	0.02
Online Pilates	41.7 ± 8.3	42.5 ± 7.7					0.10
F-F Pilates	39.8 ± 6.3	39.9 ± 6.7					0.00

F-F Pilates: face-to-face Pilates.

**Table 3 ijerph-21-01385-t003:** Functional capacity tests of volunteers allocated to control (n = 18), online Pilates (n = 16), and face-to-face Pilates (n = 15) groups.

	Pre	Post	*p* Group	*p* Time	*p* Group × Time	Power	Cohen’s *d*
Endurance Strength (repetition)							
Control	16 ± 4	18 ± 6	0.11	<0.01	0.01	0.377	0.39
Online Pilates	19 ± 5	21 ± 5					0.49
F-F Pilates	16 ± 4	23 ± 4 *					1.57
TUG (seconds)							
Control	8.0 ± 1.4	7.5 ± 0.9	0.69	<0.01	0.24	0.086	0.51
Online Pilates	7.8 ± 1.0	7.2 ± 0.9					0.61
F-F Pilates	8.1 ± 2.0	7.0 ± 0.9					0.75
Right Handgrip (kg)							
Control	30.0 ± 07.1	28.2 ± 04.2	0.38	0.34	0.10	0.158	0.30
Online Pilates	30.1 ± 07.1	31.9 ± 05.9					0.20
F-F Pilates	26.6 ± 10.0	29.6 ± 09.9					0.30
Left Handgrip (kg)							
Control	28.1 ± 7.3	25.8 ± 2.8	0.28	0.12	0.14	0.125	0.32
Online Pilates	28.4 ± 8.4	29.1 ± 6.4					0.09
F-F Pilates	25.7 ± 9.6	28.2 ± 7.1					0.29
Flexibility (cm)							
Control	15.1 ± 9.8	16.0 ± 6.3	0.92	0.01	0.13	0.212	0.10
Online Pilates	14.2 ± 7.0	15.6 ± 6.3					0.15
F-F Pilates	12.0 ± 7.1	18.4 ± 3.2					1.15
Trunk Flexion (kg)							
Control	10.1 ± 4.8	8.8 ± 2.7	0.69	0.07	<0.01	0.566	0.33
Online Pilates	09.3 ± 7.5	11.0 ± 5.6					0.26
F-F Pilates	06.6 ± 3.0	12.7 ± 6.6 *					1.17
Trunk Extension (kg)							
Control	12.7 ± 9.8	10.1 ± 6.5	0.11	0.29	0.71	0.070	0.31
Online Pilates	08.6 ± 7.0	08.1 ± 5.1					0.08
F-F Pilates	13.2 ± 8.2	12.7 ± 9.3					0.07
6 min Walk Test (m)							
Control	464 ± 67	461 ± 44	0.32	0.06	0.03	0.331	0.05
Online Pilates	467 ± 53	467 ± 42					0.00
F-F Pilates	463 ± 73	514 ± 39 **					0.87

TUG: timed up-and-go test. F-F: Face-to-face. * *p* < 0.01 in relation to pre moment. ** *p* < 0.05 in relation to pre moment.

**Table 4 ijerph-21-01385-t004:** Fasting blood test analysis and resting blood pressure of volunteers allocated to control (n = 18), online Pilates (n = 16), and face-to-face Pilates (n = 15) groups.

	Pre	Post	Cohen’s *d*	*p* Group	*p* Time	*p* Group × Time	Power
Blood Glucose (mg/dL)							
Control	95.6 ± 14.8	93.6 ± 7.4	0.16	0.77	0.52	0.24	0.052
Online Pilates	103.9 ± 32.2	103.5 ± 37	0.01				
F-F Pilates	95.7 ± 15.1	95.2 ± 7.2	0.04				
HbA1C (mg/dL)							
Control	6.0 ± 0.4	5.9 ± 0.3	0.08	0.39	0.60	0.82	0.059
Online Pilates	6.2 ± 1.3	6.3 ± 1.4	0.10				
F-F Pilates	5.8 ± 0.5	5.8 ± 0.3	0.09				
Triglycerides (mg/dL)							
Control	109.7 ± 39.3	95.9 ± 24.5	0.42	0.14	0.06	0.16	0.501
Online Pilates	133.2 ± 41.9	138.6 ± 56.4	0.10				
F-F Pilates	95.4 ± 30.3	126.9 ± 29.4	1.05				
Total Cholesterol (mg/dL)							
Control	212.7 ± 47.8	203.8 ± 62.1	0.16	0.04	0.49	0.39	0.096
Online Pilates	179.5 ± 24.4	187.2 ± 22.9	0.30				
F-F Pilates	212.1 ± 45.1	211.9 ± 40.7	0.00				
HDL-c (mg/dL)							
Control	55.3 ± 14.4	57.4 ± 17.3	0.13	0.61	0.23	0.01	0.178
Online Pilates	51.3 ± 10.8	51.0 ± 9.2	0.02				
F-F Pilates	59.6 ± 12.9	53.5 ± 9.0	0.54				
LDL-c (mg/dL)							
Control	135.5 ± 44.4	126.6 ± 2.7	0.18	0.02	0.24	0.96	0.076
Online Pilates	106.7 ± 26.2	108.9 ± 15.0	0.10				
F-F Pilates	132.4 ± 41.0	134.0 ± 41.3	0.03				
Systolic Blood Pressure (mmHg)							
Control	114 ± 13	122 ± 12 *	0.58	0.33	0.44	<0.01	0.311
Online Pilates	118 ± 13	113 ± 11	0.37				
F-F Pilates	112 ± 13	112 ± 12	0.02				
Diastolic Blood Pressure (mmHg)							
Control	75 ± 9	81 ± 8 *	0.73	0.82	0.02	0.03	0.206
Online Pilates	76 ± 9	76 ± 7	0.03				
F-F Pilates	76 ± 9	78 ± 7	0.29				

F-F: Face-to-face; HbA1C: glycated hemoglobin; HDL-c: high-density lipoprotein cholesterol; LDL-c: low-density lipoprotein cholesterol. * *p* < 0.01 in relation to pre moment.

## Data Availability

The data that supports the findings of this study are available from the corresponding author, Puga, G.M., upon reasonable request.

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
