# Peer review of "Online and Face-to-Face Mat Pilates Training for Long COVID-19 Patients: A Randomized Controlled Trial on Health Outcomes"

_ijerph, 2024, doi:10.3390/ijerph21101385_

Round 1

Reviewer 1 Report

Comments and Suggestions for Authors

The title: simplification is recommended, it is difficult to interpret, with unnecessary punctuation.

Abstract: adequate

Introduction

The first section of the introduction is adequate, the paragraph presenting the Pilates method is inadequate, non-specific, general, one sentence is closely related to the topic: 'improves 61 physical capacities exacerbated by Long COVID [18–22]'.

The introduction is incomplete, it does not explain why Pilates was studied in middle-aged patients.

Methods

The trial was not registered with ClinicalTrial.gov.

Participants

It is not really understandable, if the formulated goal was aimed at middle-aged participants, why was the program announced for people between the ages of 18 and 60? Please explain!

Accurate formulation of the inclusion and exclusion criteria is essential for all research. In this case, both need to be supplemented.

The type of work the participants performed during the study was not measured.

Outcomes

The authors do not describe the exact implementation rules of circumference measurement.

References are incomplete or inaccurate in this section.

'The test evaluators were unaware of which group each participant belonged to.' I recommend moving this sentence to the introductory part of the outcomes, it is an essential element. (line 122)

Intervention and control

In this section, there is no reference to how did you find out about the co-morbidities of the participants?

How was the condition of the patients checked before starting the exercise program? (Present condition or present symptoms, participants are able to perform the exercises)

Was there an unexpected event, according to which protocol did the group leaders act?

What protocol was used to determine that a 45-second interval between exercises is necessary for muscle recovery?

The Table 1. 'Twelve-week Mat Pilates training progression' is completely unnecessary, here is no extra information, please delete.

If the control group did stretching exercises, Pilates was compared to unsupervised stretching. Please reconsider!

What was the main reason for the absences?

The paragraph from line 175 to line 187 contains information irrelevant to the research.

Results

The data in the flowchart are difficult to follow and incomprehensible. (This is part of the method section. Not a result.)

Taking medication and assessing co-morbidities are part of the presentation of the participants in the method section. Please correct it!

How did the participants confirm that they had a covid infection? Who made the long covid diagnosis?

It is not necessary to describe the categories of obesity in Table 2, the BMI provides enough information.

How could the participants be inactive during the Pilates program? Please explain it!

The tables and a figure are adequate. (Significant results should be highlighted to make it easier for the reader to follow the results.)

Discussion

The discussion part is a bit long-winded; the authors could make this section more interesting by keeping the essential elements.

Irrelevant. (Education program dealing with heart and joint problems. Lines 266-268)

The 'body aches' is irrelevant. It was not the purpose of the study. (Lines 272)

Conclusions are adequate.

Reference

Almost half of the literature references are older than 5 years, 3 cannot be identified for the reader.

Supplementary Material

The supplementary material is too long (10 pages), the authors summarize the substantive/relevant part in the article when describing the intervention.

Author Response

Ref: ijerph-3253346

Title: Mat Pilates for Long Covid-19: Online vs. Face-to-Face - A Randomized Controlled Trial on Health Outcomes

Dear, Paul B. Tchounwou, Editor-in-chief of International Journal of Environmental Research and Public Health, and Dana Sopoian Assigned Editor,

We would like to thank you for the opportunity to revise and resubmit our manuscript.  The reviewers’ comments were extremely helpful for improving our paper. Please find below our answers to each comment. We performed a revision of the entire manuscript according to the reviewers’ suggestions which will hopefully improve the clarity and readability of the manuscript. To make the changes more visible, all edits to the manuscript were highlighted in yellow.

Reviewer 1:

Comments 1: The title: simplification is recommended, it is difficult to interpret, with unnecessary punctuation.

Response 1: We appreciate the suggestion. We modified the title as mentioned.

Comments 2: Introduction: The first section of the introduction is adequate, the paragraph presenting the Pilates method is inadequate, non-specific, general, one sentence is closely related to the topic: 'improves 61 physical capacities exacerbated by Long COVID [18–22]'.

Response 2: Thanks for the comment. We changed this section for better understanding – line 57 to 62.

Comments 3: The trial was not registered with ClinicalTrial.gov.

Response 3: This study was registered with the Brazilian Registry of Clinical Trials (ReBEC) - UNT: U1111-1292-5013, as suggested in ClinicalTrials.gov web page to do first. Both platforms share the same objective.

Comments 4: It is not really understandable, if the formulated goal was aimed at middle-aged participants, why was the program announced for people between the ages of 18 and 60? Please explain!

Response 4: The target population of the study consisted of adults aged between 18 and 60 years, however, only middle-aged individuals participated entire study. Some participants younger than 40 years signed up to participate in the study, but they did not completed the initials evaluation and gave up the study. The term "middle-aged" was replaced to "adults."

Comments 5: The type of work the participants performed during the study was not measured.

Response 5: We are not sure if we understand the comment, but if you are talking about the total work of the exercise (multiplying force by displacement) ,  It was not possible to measured it in the study. Although we do measured the heart rate and rate of perceived exertion during the entire intervention, and it may help us to estimate the exercise intensity and total workload of the exercise.

But if you are talking about the job of the participants, mostly of then worked in office and none of them had manual labor that need high physical fitness to perform it. We do apologized for the misunderstanding.

Comments 6: The authors do not describe the exact implementation rules of circumference measurement.

Response 6: Thanks for the comment. We changed this section for better understanding. We described the circumference measurement in the text – 2.9 session, lines 136 to 137

Comments 7: 'The test evaluators were unaware of which group each participant belonged to.' I recommend moving this sentence to the introductory part of the outcomes, it is an essential element. (line 122)

Response 7: Thanks for the comment. We changed the sentence "The test evaluators were unaware of which group each participant belonged to." to the beginning of the outcomes section – line 103.

Comments 8: In this section, there is no reference to how did you find out about the co-morbidities of the participants?

Response 8: The participants reported the comorbidities in the anamnesis form and we reported it based on the physical exams that all participants of the study presented to participate in the study, and also based on the medication that they were in use. This information were collected to better characterize the participants. We add this information in the first paragraph of results session lines 234 to 236.

Comments 9: How was the condition of the patients checked before starting the exercise program? (Present condition or present symptoms, participants are able to perform the exercises)

Response 9: All participants presented physical exams made by their own physician, attesting that they could practice physical exercises. It was also a recommendation for all participants to practice exercises to improve their fatigue due long COVID condition.

Comments 10: Was there an unexpected event, according to which protocol did the group leaders act?

Response 10: Thank you for asking this. No, there were no unexpected event during the entire study, that could change the protocol or interventions. All participant of the study were in a fatigue condition due long COVID, but none of them needed medical assistant during the study.

Comments 11: What protocol was used to determine that a 45-second interval between exercises is necessary for muscle recovery?

Response 11: Thank you for the comment. There is no pattern in the rest interval duration in Mat Pilates exercise training. Nevertheless the original method of Pilates exercise uses from 30 to 90 seconds of interval duration between exercises. We also based this interval with resistance training patterns when then use circuit training. Previously studies using Mat Pilates training from our group used this interval and works fine to this exercise modality.

- Tavares JB, Batista JP, Costa JG, Gonçalves LF, de Souza TCF, Mariano IM, Amaral AL, Rodrigues ML, Puga GM. Comparison of Mat Pilates training-induced changes on climateric symptoms in hypertensive and normotensive postmenopausal women. J Bodyw Mov Ther. 2022 Oct;32:102-109. doi: 10.1016/j.jbmt.2022.05.008.

- Amaral AL, Batista JP, Mariano IM, Gonçalves LF, Tavares JB, de Souza AV, Caixeta DC, Teixeira RR, de Oliveira EP, Espindola FS, Puga GM. Redox Status of Postmenopausal Women with Single or Multiple Cardiometabolic Diseases Has a Similar Response to Mat Pilates Training. Antioxidants (Basel). 2022 Jul 26;11(8):1445. doi: 10.3390/antiox11081445. 

- Tavares JB, Batista JP, Costa JG, Gonçalves LF, de Souza TCF, Mariano IM, Amaral AL, Rodrigues ML, Puga GM. Comparison of Mat Pilates training-induced changes on climateric symptoms in hypertensive and normotensive postmenopausal women. J Bodyw Mov Ther. 2022 Oct;32:102-109. doi: 10.1016/j.jbmt.2022.05.008.

Comments 12: The Table 1. 'Twelve-week Mat Pilates training progression' is completely unnecessary, here is no extra information, please delete.

Response 12: Thank you for the comment. We do believe that for training prescription and study reproduction, the information that Table 1 shows is important. It illustrates the progression of the training, which is not described elsewhere in the text. So we add this information in the text (lines 178 to 180) and excluded the table as suggested.

Comments 13: If the control group did stretching exercises, Pilates was compared to unsupervised stretching. Please reconsider!

Response 13: Thank you for the comment. The exercise research guidelines and ethical committee usually recommend that when you have an intervention group with exercise prescription, and a control group with no intervention, is important to offer some activity to the control group. This activity will not change the energy expenditure, physiological responses or muscle action at the same way as the exercise intervention. But it is important to expose the participants to the same environment as the intervention group. The stretching that was performed by the control group was very light and with no physical stress as the Pilates groups. So we do used unsupervised stretching in control group, but we controlled and can guarantee that this activity was very low and did not have potential physiological adaptations as the exercise group.

Comments 14: What was the main reason for the absences?

Response 14: The main reasons for the absences were scheduling conflicts or health issues not related to the study.

Comments 15: The paragraph from line 175 to line 187 contains information irrelevant to the research

Response 15: We do think that the information described in the paragraph from line 175 to line 187 is important, because it explains some information that helps the reproducibility of the study protocol. This information also helps to understand the way that online and face-to-face Pilates was performed.

Comments 16: The data in the flowchart are difficult to follow and incomprehensible. (This is part of the method section. Not a result.).

Response 16: The flowchart followed the model of the Consolidated Standards of Reporting Trials (CONSORT). We change some information for better understanding. We included this figure in the Methods section as suggested.

Comments 17: Taking medication and assessing co-morbidities are part of the presentation of the participants in the method section. Please correct it!

Response 17: Thank you for the suggestion. We add this information in the methods session, “participants” – page 3.

Comments 18: How did the participants confirm that they had a covid infection? Who made the long covid diagnosis?

Response 18: COVID-19 infection was confirmed by viral tests including nucleic acid amplification tests, PCR testes or antigen tests. All tests were performed in a public or private health care location and confirmed with the symptoms by a physician. The patients could only participate in the study after proving that had a positive COVID 19 test with presence of symptoms for at least 3 months. We add this information in the methods session lines 94 to 100.

Comments 19: It is not necessary to describe the categories of obesity in Table 2, the BMI provides enough information.

Response 19: Thank you for the suggestion. The classification of Body Mass Index (BMI) was removed from Table 1 (the number of tables changed).

Comments 20: How could the participants be inactive during the Pilates program? Please explain it!

Response 20: Some of participants (3 in the control and 1 in the F-F group) were inactive before the Pilates program. Not during. It’s important to know that all participants were familiarized with the Pilates exercises methods before the study began. Table 1 shows the Baseline general characteristics of volunteers, so it was before the Mat Pilates exercise training.

Comments 21: The tables and a figure are adequate. (Significant results should be highlighted to make it easier for the reader to follow the results.)

Response 21: Thank you for the comment. We used the symbols * to indicate significant results in the tables. We also highlight it in bold.

Comments 22: The discussion part is a bit long-winded; the authors could make this section more interesting by keeping the essential elements.

Response 22: We appreciate the comment. We adjusted the text as suggested.

Comments 23: Irrelevant. (Education program dealing with heart and joint problems. Lines 266-268)

Response 23: We appreciate the comment. We adjusted the text as suggested.

Comments 24: The 'body aches' is irrelevant. It was not the purpose of the study. (Lines 272)

Response 24: We appreciate the comment. We adjusted the text as suggested.

Comments 25: Almost half of the literature references are older than 5 years, 3 cannot be identified for the reader.

Response 25: We appreciate the comment. We change the references as suggested. Some Pilates references are older than 5 years because there are no so many studies about Pilates and the cardiometabolic parameters and functional capacity that we avaluated.

Comments 26: The supplementary material is too long (10 pages), the authors summarize the substantive/relevant part in the article when describing the intervention.

Response 26: We appreciate the comment. We made some changes to improve the document. However we cannot exclude it because is extremely important for replicating the Mat Pilates exercises training. That is why we add it as supplementary material.

Reviewer 2 Report

Comments and Suggestions for Authors

Article: MAT PILATES FOR LONG COVID-19: ONLINE VS. FACE- 2

TO-FACE – A RANDOMIZED CONTROLLED TRIAL ON 3

HEALTH OUTCOMES

Aspects to improve:

1. In the summary it is important that it is structured as follows: Introduction, Objective, methodology, results, discussion and conclusions, which allows the reader to have a better understanding regarding the study.

2. Methodology should describe the process of a controlled clinical trial, type of research, population (inclusion and exclusion criteria), techniques and instruments (that mediate each of them), type of analysis, the intervention performed (control group and intervention) that will help to have a logical sequence of the development of the study.

3. Develop the conclusions obtained in the study and incorporate a subsection on the limitations of the study and new lines of research.

Author Response

Ref: ijerph-3253346

Title: Mat Pilates for Long Covid-19: Online vs. Face-to-Face - A Randomized Controlled Trial on Health Outcomes

Dear, Paul B. Tchounwou, Editor-in-chief of International Journal of Environmental Research and Public Health, and Dana Sopoian Assigned Editor,

We would like to thank you for the opportunity to revise and resubmit our manuscript.  The reviewers’ comments were extremely helpful for improving our paper. Please find below our answers to each comment. We performed a revision of the entire manuscript according to the reviewers’ suggestions which will hopefully improve the clarity and readability of the manuscript. To make the changes more visible, all edits to the manuscript were highlighted in yellow.

Comments 1: In the summary it is important that it is structured as follows: Introduction, Objective, methodology, results, discussion and conclusions, which allows the reader to have a better understanding regarding the study.

Response 1: We appreciate the comment. We used the journal’s template and followed the instructions for authors to prepare the entire manuscript. The objective of the study is presented in the final paragraph of the introduction.

Comments 2: Methodology should describe the process of a controlled clinical trial, type of research, population (inclusion and exclusion criteria), techniques and instruments (that mediate each of them), type of analysis, the intervention performed (control group and intervention) that will help to have a logical sequence of the development of the study.

Response 2: Thank you for the comment. We addresses this information in the Materials and Methods session. We highlighted it in yellow and modified for better understanding. We used the CONSORT check list to describe all this information. We also divided the information of this session in topics for better understanding.

Comments 3: Develop the conclusions obtained in the study and incorporate a subsection on the limitations of the study and new lines of research.

Response 3: We appreciate the comment. The limitations of this study and suggestions for future research were included in the conclusion.